# *SC-GROG* followed *by L+S* reconstruction with multiple sparsity constraints for accelerated Golden-angle-radial DCE-MRI

**Faisal Najeeb**[1], **Kashif Amjad**[2]*, **Irfan Ullah**[1], **Hammad Omer**[1]

**1** MIPRG Research Group, Electrical and Computer Engineering Department, COMSATs University Islamabad, Islamabad, Pakistan, **2** College of Computer Engineering & Science, Prince Mohammad Bin Fahd University, Khobar, Saudi Arabia

* kamjad@pmu.edu.sa

**Data Availability Statement:** The proposed method is tested on two different Human abdomen free-breathing Golden-angle-radial DCE-MRI datasets available at https://cai2r.net/resources/

## Abstract

The GRASP (Golden-angle-radial Sparse Parallel MRI) is a contemporary method for reconstructing dynamic contrast-enhanced magnetic resonance imaging (DCE-MRI). This method combines the temporal incoherence of stack-of-stars Golden-angle-radial sampling pattern and acceleration capability of parallel MRI (PI) and compressed sensing (CS) for highly accelerated free-breathing DCE-MRI reconstruction. GRASP uses Temporal Total Variation (TV) norm as a sparsity transform to promote sparsity among multi-coil MRI data and Nonlinear Conjugate Gradient (NL-CG) algorithm to obtain an optimal solution. Additionally, GRASP uses NUFFT gridding to map Golden-angle-radial data to Cartesian grid before NL-CG based CS reconstruction. However, major limitations of GRASP include the temporal averaging effect due to Temporal TV, leading to a degradation in the dynamic contrast of DCE-MRI, and a high computational burden/reconstruction time due to repeated NUFFT gridding/degridding in NL-CG reconstruction. This paper introduces a novel approach to address limitations in GRASP reconstruction technique for free-breathing DCE-MRI. The proposed method combines SC-GROG gridding with low-rank plus sparse (L+S) reconstruction using multiple sparsity constraints for accelerated Golden-angle-radial DCE-MRI with improved temporal resolution and dynamic contrast. Monotone FISTA with variable acceleration (MFISTA-VA) is used to optimize the L+S optimization problem. Further, SC-GROG gridding is used to map Golden-angle radial data to Cartesian grid before *L+S* reconstruction. The proposed method is tested on two different 3T free-breathing in-vivo DCE-MRI datasets. Reconstruction results of the proposed method are evaluated by using: (i) convergence error, (ii) peak and mean values of arterial signal intensity in the selected region of interest (ROI) of DCE MR Images, and (iii) reconstruction time. Results show that the proposed method provides significant improvements in the reconstruction time and dynamic contrast than the conventional Golden-angle-radial DCE-MRI reconstruction techniques (i.e., GRASP, XD-GRASP). Furthermore, convergence analysis shows that integration of MFISTA-VA in *L+S* reconstruction provides faster convergence compared to conventional *L+S* reconstruction.

#reconstruction-code. We downloaded Dataset 1 by a download link request at https://cai2r.net/resources/xd-grasp-matlab-code/ and dataset 2 by a download link request at https://cai2r.net/resources/racer-grasp-reconstruction-code/.

**Funding:** The author(s) received no specific funding for this work.

**Competing interests:** The authors have declared that no competing interests exist.

## 1. Introduction

Dynamic Contrast-Enhanced MRI (DCE-MRI) is widely used to study the characteristics of microvasculature in many physiological and pathological instances [1–3]. Uptake and washout of the contrast agent (CA) in DCE-MRI takes several minutes. The patient feels distressed due to the long scan process while physiological movement (e.g., respiratory, and cardiac motion) degrades the dynamic contrast of MR image [2–4]. To generate good quality DCE MR images, signal intensity variation due to contrast agent and the respiratory motion must be distinguished from each other before image reconstruction [3].

Many efforts have been made in the recent past to accelerate DCE MR image reconstruction with a high spatio-temporal resolution [5–12]. GRASP [1] proposed by Feng et al. provides rapid imaging with improved temporal resolution compared to conventional compressed sensing (CS) and parallel MRI (PI) reconstruction techniques [13, 14]. In GRASP, data are continuously acquired using Golden-angle-radial trajectory and the contrast-enhanced images are reconstructed retrospectively with feasible spatio-temporal resolutions to meet different clinical requirements [1]. However, high computation burden due to NUFFT (Fessler) gridding and respiratory motion artefacts limits its clinical application. XD-GRASP is another technique in which respiratory-motion signal is detected directly from the acquired free-breathing Golden-angle-radial $k$-space data using the projection approach [9]. In XD-GRASP multiple motion state DCE MR images with different temporal resolution were reconstructed using Non-Linear Conjugate Gradient (NL-CG) reconstruction. Reconstruction of multiple motion state images and repeated gridding/degridding in the NL-CG reconstruction for data consistency further extends the reconstruction period.

Besides the reconstruction efficiency, dynamic contrast performance also plays a vital role in clinical diagnosis. GRASP based techniques [1, 2, 6, 9, 13, 15] typically employ temporal TV as a sparsity transform to exploit sparsity among multi-coil DCE-MRI data. Temporal TV can suppress all the temporal variations, including temporal incoherent under-sampling artifacts, it also leads to the temporal averaging effect, which degrades the ultimate dynamic contrast of DCE-MRI [16].

Although GRASP-based reconstruction techniques for DCE-MRI [1, 6, 9, 13, 15] demonstrate promising imaging performance, however, there are two major challenges associated with these techniques: (i) temporal averaging effect due to Temporal TV, which degrades the ultimate dynamic contrast of DCE-MRI and (ii) computationally expensive NL-CG reconstruction which requires repeated NUFFT gridding/degridding to maintain data consistency during iterative CS reconstruction [10, 12].

Low-rank matrix completion which is an extended version of CS, recovers the missing or corrupted entries of a matrix under low-rank and incoherent conditions [17–23]. Application of $L+S$ reconstruction (discussed in section 2.2) in DCE-MRI improves contrast visualization due to background suppression in the S component, which is useful for the detection of low enhanced regions. Standard $L+S$ reconstruction uses singular value thresholding (SVT) for matrix completion and iterative soft-thresholding algorithm (ISTA) [19] for sparse representation. ISTA is a proximal-gradient based method, which was proposed almost a decade ago [24, 25]. There are different variants of ISTA e.g., Fast Iterative soft-thresholding Algorithm (FISTA), accelerated FISTA (AFISTA) [26], Optimized ISTA (OISTA) [27, 28] and optimized gradient method (OGM) [29]. These algorithms provide faster convergence compared to ISTA. Monotone FISTA with variable acceleration (MFISTA-VA) [30] has been recently proposed which provides faster convergence compared to contemporary proximal-gradient based ISTA methods [20, 24–26, 28, 29].

This work aims to improve the dynamic contrast and reduce the reconstruction time of CS-based image reconstruction problems in DCE-MRI. In this paper, we propose SC-GROG followed by MFISTA-VA based $L+S$ reconstruction using multiple sparsity constraints for

Golden-angle-radial DCE-MRI with improved temporal resolution and dynamic contrast. Conventional $L+S$ reconstruction method uses ISTA to obtain the optimal solution [19]. This paper uses Monotone FISTA with variable acceleration (MFISTA-VA) to solve the $L+S$ optimization problem; the results show improved convergence than conventional $L+S$ reconstruction. This work also uses SC-GROG to convert Golden-angle radial data to Cartesian grid before $L+S$ reconstruction. The rest of the paper is organized as follows: Section 2 provides an overview about iterative thresholding reconstruction, conventional $L+S$ decomposition model and SC-GROG gridding. Section 3 describes the proposed method, section 4 provides details about the datasets and simulation environment in our experiments, and section 5 presents the reconstruction results.

## 2. Theory

### 2.1 CS based MR image reconstruction

CS based MR image reconstruction can be considered as an optimization problem defined over the complex vectors $x, y \in C^n$; that is vectors with complex components. Let us consider the problem defined as:

$$\Psi(x) = f(x) + h(x) \tag{2.1}$$

Here $h(x)$ represents the sparsity term for $x$, $f:C^n \to \mathbb{R}$ is convex and continuously differentiable. It also satisfies $\|\nabla f(x) - \nabla f(y)\| < \mathcal{L}\|x - y\|$ for some constant $\mathcal{L} \in \mathbb{R}$. $\mathcal{L}$, represents the Lipschitz constant, $\| \|$ denotes Euclidean norm, and $\nabla$ represents the gradient operator. In Eq (2.1), $h: C^n \to \mathbb{R}$ is also convex but may not be smooth.

In the context of CS-MRI based optimization problems, reconstruction of an MR image x, given the partially acquired $k$-space data $d$, is formulated as [31]:

$$\Psi(x) = \underset{x}{argmin} \; \frac{1}{2}\|E(x) - d\|_2^2 + \lambda\|T(x)\|_1 \tag{2.2}$$

In Eq (2.2), the vector $x \in C^n$ represents the MR image to be reconstructed and vector $d \in C^m$ represents the under-sampled k-space data (partially acquired from the scanner). $E$ is the encoding operator which contains Fourier sampling and coil-maps information. Similarly, $\lambda$ is the regularization parameter and $T$ represents the sparsity transform, which could be temporal $TV$ or temporal $FFT$ [16].

The reconstruction time, i.e., the time required to minimize Eq (2.2), is extremely important in CS-based MR applications, particularly for non-Cartesian MR image reconstruction [16, 20, 31]. In such problems having ill-conditioned system matrices, fast algorithms are required to solve the non-differentiability of the sparsity term $h(x)$. The proximal-gradient methods can deal with non-differentiability of $h(x)$. In these methods, a Proximal-gradient step (with step size $\frac{1}{w}$) is defined as follows [27]:

$$P_w(y) := \underset{h,\, w}{Prox}\left(y - \frac{1}{w}\nabla f(y)\right) \tag{2.3}$$

where

$$\underset{h,\, w}{Prox}(x) := \underset{z \in c^n}{\arg min}\left\{h(z) + \frac{w}{2}\|z - x\|_2^2\right\} \tag{2.4}$$

is the proximal operator [27–29] of $h$ for parameter $w \in \mathbb{R}$.

**2.1.1 Monotone FISTA with Variable Acceleration (MFISTA-VA).** Iterative shrinkage-thresholding/soft-thresholding (ISTA), Fast iterative shrinkage-thresholding (FISTA) and monotone FISTA are basic proximal-gradient based methods which were proposed almost a decade ago [24–26, 32]. Let $w \geq \mathcal{L}$ (Lipschitz constant), then a major condition to converge faster for ISTA and FISTA based proximal-gradient methods is:

$$\Psi(x_k) - \Psi(\mathrm{x}^*) \leq \frac{2\,w\|x_0 - x^*\|_2^2}{(k+1)^2} \tag{2.5}$$

Here $x^*$ represents the minimizer of the $\Psi$, $x_k$ denotes the $k_{th}$ iteration in M-FISTA with the constant parameter $w$ as the proximal operator step. To satisfy the convergence condition in Eq (2.5), FISTA and MFISTA combine a proximal-gradient step (step-4 in Table 1) with a momentum step suggested by Nesterov et al. in [28]. MFISTA-VA [30] is a recently proposed method which improves convergence of conventional ISTA and FISTA based algorithms [26, 27, 33].

Table 1 shows the algorithm for MFISTA-VA. ISTA uses step size, $z_k = 1$, while both MFISTA and MFISTA-VA have step size $z_k = P_{w_k}(y_k)$ (step 3 in Table 1). This step arises by minimizing the quadratic surrogate function $Q_L(z, y)$ (used in step 6 of Table 1). Further, in MFISTA-VA, $w$ in right hand side of Eq (2.5) was replaced by $\frac{w_k}{\eta_k}$, where $w_k$ may be less than $\mathcal{L}$ ((Lipschitz constant). $\eta_k$ is calculated during each iterative step as:

$$\eta_k = 1 + 2\frac{Q_{w_k}(z_k - x_k) - \Psi(x_k)}{w_k\|y_k - z_k\|^2} \tag{2.6}$$

This $\eta_k$ is used to calculate the multiplier of proximal gradient step (an extra term in momentum formula) also called variable acceleration (VA). This multiplier increases convergence of MFISTA-VA compared to FISTA which uses constant step size without multiplier and ISTA which uses unit step size (step size = 1).

## 2.2 MR image reconstruction using robust matrix decomposition

Otazo et.al. proposed the $L+S$ reconstruction model that integrates principal component analysis (PCA), PI and CS to reconstruct artefact-free MR images from the under-sampled dynamic MRI data [19]. This method aims to decompose a matrix $M$ into a low rank $L$ component (containing few non-zero singular values) and a sparse $S$ component (containing few nonzero entries). $L+S$ decomposition is mathematically formulated by solving convex

**Table 1. Algorithm for MFISTA-VA [30].**

**Initialize**
1: $y_1 = x_0$, $t_1 = 1$
2: **For** $k = 1$ *to* $N$ *do*
3: $z_k = P_{w_k}(y_k)$
% Step size during each iteration
4: $x_k = \underset{x \in \{z_k, x_{k-1}\}}{\arg\min}\ \Psi(x)$
5: $t_{k+1} = (1 + \sqrt{1 + 4(t_k)^2})/2$
6: $\eta_k = 1 + 2\frac{Q_{w_k}(z_k - x_k) - \Psi(x_k)}{w_k\|y_k - z_k\|^2}$
7: $y_{k+1} = x + \frac{t_{k-1}}{t_{k+1}}(x_k - x_{k-1}) + \frac{t_k}{t_{k+1}}(z_k - x_k) + \frac{t_k}{t_{k+1}}(\eta_k - 1)(z_k - y_k)$
8: **end for**

optimization problem as:

$$\min\|L\|_* + \lambda_T\|S\|_1 \, s.t \, M = L + S \tag{2.7}$$

Here, the sparse matrix S models the grossly corrupted noise/outlier information which is assumed to be a very small fraction of the acquired MRI data. $\lambda_T$ represents regularization parameters for S component. To reconstruct multi-coil under-sampled dynamic MRI data, the L+S decomposition in Eq (2.7) could be formulated by using convex optimization as [19]:

$$\underset{L,S}{argmin} \, \frac{1}{2}\|E(L+S) - d\|_2^2 + \lambda_L\|L\|_* + \lambda_T\|TS\|_1 \tag{2.8}$$

where $d$ is the under-sampled $k$-space data acquired from the MRI scanner, $\|\,\|_*$ being the nuclear norm which is used to exploit the low-rank component '$L$'. $\|L\|_*$ is defined as $\|L\|_* = \sum_{i=1}^n \sigma_i$ where $\sigma_1, \sigma_2. \ldots .,\sigma_n$ are the singular values of the low rank ($L$) matrix and $n$ represents the rank of $L$. $\lambda_L$ is the regularization parameters for the $L$ components. The rest of the parameters are the same as defined in Eq (2.2). $L+S$ reconstruction uses iterative soft thresholding algorithm (ISTA) for sparse representation and singular value thresholding (SVT) for the matrix completion [19, 31].

Table 2 provides the framework of $L+S$ reconstruction model [19] in which background component $L$ is obtained by applying thresholding on the singular values of the data matrix $d$ while the dynamic component $S$ is also obtained by subtracting the low rank ($L$) component from the data matrix $M$.

In $L+S$ decomposition, SVT is used for soft thresholding the singular values of the low rank components in each iteration as $L_k = SVT(Y_{k-1} - S_{k-1})$. Simlarly ISTA is used for soft thresholding of the sparsity constraint $S_k = T^{-1}(\wedge_{\lambda_T}(T(Y_{k-1} - L_{k-1})))$ during each iteration. The image series is recovered as $Y_k = L_k + S_k - E^*(E(L_k + S_k - d))$. To maintain data consistency (DC) during image reconstruction in every kth iteration, the residual signal in the k-space $E^*(E(L_k + S_k - d))$ is subtracted from $L_k+S_k$.

**Table 2. Standard *L+S* reconstruction framework using ISTA [19].**

| *L+S* Reconstruction using ISTA. |
|---|
| **Input** |
| d: multi-coil undersampled *k*-data |
| E: multi-coil encoding operator (NUFFT gridding and coil maps information) |
| T: temporal TV based sparsity transform |
| $\lambda_L$: singular-value threshold |
| $\lambda_T$: threshold for temporal TV |
| **Initialize** |
| 1: $M_0 = E*d, S_0 = 0, L_1 = 1$ |
| 2: **for** $k = 1$ to $N$ do |
| % Singular value thresholding |
| 3: $L_k = SVT_{\lambda_L}(M_k - S_{k-1})$ |
| % Soft thresholding in the transform domain |
| 4: $S_k = T^{-1}(\wedge_{\lambda_T}(T(M_k - L_{k-1})))$ |
| % Data consistency |
| 5: $Y_k = L_k + S_k - E^*(E(L_k + S_k - d))$ |
| 6: **end for** |
| **Output** $Y_k$ |

Convergence properties of L+S algorithm [19] could be analyzed by considering it as a special case of the proximal gradient method for solving convex optimization in Eq (2.7) as:

$$\min f(x) + h(x) \tag{2.9}$$

Here '$f$' represents quadratic term in Eq (2.7) which is convex and smooth. Similarly, '$h$' represents sparsity term (which is the sum of the nuclear and $l1$ norms in Eq (2.7)) which may be convex but not necessarily smooth. For L+S reconstruction-based optimization problem, proximal gradient method takes the form as:

$$x_k = prox_h(x_{k-1} - t_k \nabla f(x_{k-1})) \tag{2.10}$$

Here, $t_k$ shows a step size in every iteration and $prox_h$ is the proximity function for $h$ which can be described as follows:

$$prox_h(y) = \underset{x}{argmin} \; \frac{1}{2}\|y - x\|_2^2 + h(x) \tag{2.11}$$

For a constant step size, Eq (2.8) can be written as:

$$L_k = SVT_{\lambda L}(L_{k-1} - t_k E^*(E(L_k + S_k) - d)) \tag{2.12}$$

$$S_k = T^*[\wedge_{\lambda T}(T[(S_{k-1} - t_k E^*(E(L_k + S_k) - d))] \tag{2.13}$$

Here '$t_k$' represents the step size in k$^{th}$ iteration. For constant value of $t$, Eqs 2.12 and 2.13 are equivalent to the iterations provided in Table 2 (*L+S* reconstruction using ISTA). Standard *L+S* reconstruction uses ISTA [31] to solve the optimization problem given in Eq (2.8). ISTA is very basic proximal-gradient based method which was proposed almost a decade ago [24, 25]. It uses constant step size. In this paper, we propose MFISTA-VA [30] based optimized L+S reconstruction. MFISTA-VA uses variable step size and a multiplier (explained in Table 1) which improves the convergence efficiency of L+S decomposition model [19].

## 3. Proposed method

This paper proposes optimized *L+S* reconstruction for accelerated Golden-angle-radial DCE-MRI. The main objective of the proposed work is to reconstruct artefact free DCE-MR images from highly under-sampled Golden-angle-radial data acquired during free-breathing. A Flowchart of the proposed method is shown in Fig 1. Initially coil sensitivity maps are estimated in the image domain, where SC-GROG [34] is used on the composite dataset before data sorting (temporal frame subdivision) to get the multi-coil reference images. Coil sensitivity maps are estimated from these reference images by using Walsh method [35]. In the 2$^{nd}$ step, continuously acquired radial k-space data are sorted into different respiratory states which span from expiration (top) to inspiration (bottom) by using a respiratory motion signal. This respiratory motion signal was estimated from the acquired data by using the projection approach [12]. In the 3$^{rd}$ step, SC-GROG-gridding is applied to interpolate the Golden-angle-radial data of each motion state to a Cartesian matrix:

$$Y_c = G(Y_r) \tag{3.1}$$

Where $Y_r$ shows the Golden-angle radial data acquired from the scanner, $G$ is the GROG operator and $Y_c$ is the corresponding multi-coil Cartesian $k$-data. Finally, MFISTA-VA based optimized *L+S* reconstruction (Table 3) is used to get the solution image. The proposed method is

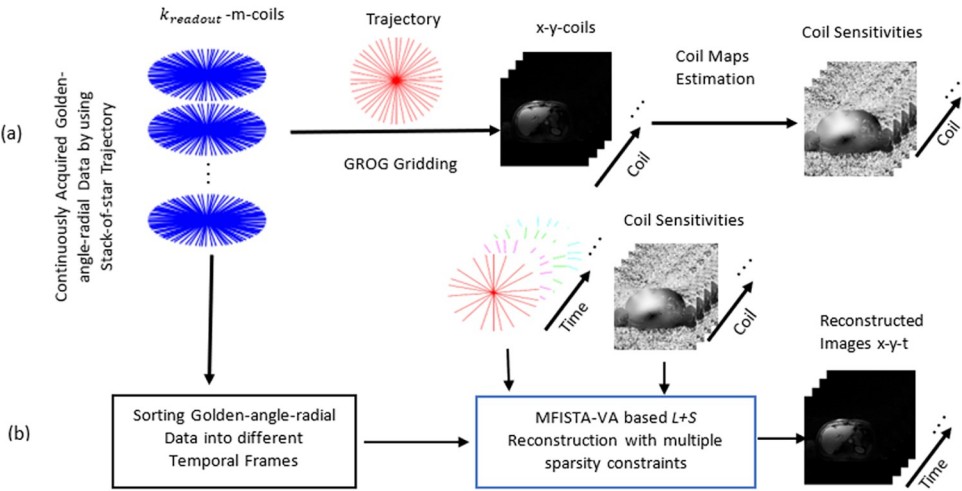

**Fig 1.** Flow-graph of the proposed method: (a) coil sensitivity maps estimation in image domain, where SC-GROG is used on the composite data set before data sorting (temporal frame subdivision) to get the multi-coil reference images. Walsh method is used to estimate coil sensitivity maps from these reference images; (b) Reconstruction of DCE MR image series, where continuously acquired *k*-data are sorted into different under-sampled dynamic time series according to temporal order. Then, respiratory motion signal is used to bin each contrast phase into different respiratory motion states/phases starting from the end-expiration to end-inspiration in a way that each motion state contains equal radial measurements. Afterwards, SC-GROG gridding is used to interpolate the radially sampled *k*-data onto a Cartesian plane. Finally, MFISTA-VA based optimized *L+S* reconstruction with multiple sparsity constraints (Eq 3.2) is applied on this GROG gridded Cartesian data to get the solution images.

**Table 3. Framework of the proposed method.**

| *Optimized L+S* **using MFISTA-VA** |
| --- |

**Input**
$Y_c$: multi-coil under-sampled Cartesian k-t data obtained after SC-GROG gridding
E: multi-coil k-t encoding operator
T: Temporal TV based sparsity transforms
F: Temporal FFT based sparsity Transform
$\lambda_L$: singular-value threshold
$\lambda_T$: sparsity threshold for temporal TV
$\lambda_F$: sparsity threshold for temporal FFT
**Initialize**
1. $d = Y_c, Y_0 = E*d, R_1 = M_0, S_0 = 0, t_1 = 1$
2. **for** $k = 1$ to $N$ do
3. $L_k = SVT(R_k - S_{k-1})$
% Soft thresholding in temporal TV domain
4. $S_{Tk} = T^{-1}(\wedge_{\lambda T}(T(R_k - L_{k-1})))$
% Soft thresholding in temporal FFT domain
5. $S_{Fk} = T^{-1}(\wedge_{\lambda F}(F(R_k - L_{k-1})))$
% Linear combination of multiple sparsity ($S_{Tk}$ and $S_{Fk}$)
6. $S_k = (S_{Tk} + S_{Fk})/2$
% Proximal gradient with step size w
7. $z_k = P_{w_k}(y_k)$
% Calculate multiplier of Proximal gradient
8. $\eta_k = 1 + 2\frac{Q_{w_k}(z_k - x_k) - h(x_k)}{w_k \|y_k - z_k\|^2}$
% Data Consistency Operation
9. $Y_k = L_k + S_k - E^*(E(L_k + S_k - d))$
% Iteration for the step-size
10. $t_{k+1} = (1 + \sqrt{1 + 4(t_k)^2})/2$
% Specific linear combination as the next input
11. $R_{k+1} = Y_k + \left(\frac{t_{k-1}}{t_{k+1}}\right)(Y_k - Y_{k-1}) + \frac{t_k}{t_{k+1}}(z_k - Y_k) + \frac{t_k}{t_{k+1}}(\eta_k - 1)(z_k - Y_k)$
12. **end for**
**Output:** *L, S*

mathematically formulated as:

$$min_{L,S} \frac{1}{2}\|E(L+S) - Y_c\|_2 + \lambda_L\|L\|_* + \lambda_T\|TS\|_1 + \lambda_F\|FS\|_1 \qquad (3.2)$$

Where $Y_c$ represents the undersampled SC-GROG gridded Cartesian $K$-space data. F is temporal FFT based sparsity Transform and $Y_F$ is the sparsity threshold for temporal $FFT$. In Eq (3.2), T represents the temporal TV based sparsity transform. This temporal TV transform applies finite-differences operator along the columns of y-t matrix S. Temporal FFT is used to convert the images series matrix from the y-t domain to the temporal-frequency (y-f) domain by applying an FFT operator along the temporal dimension. In the y-f domain, a few low-frequency components typically occupy most of the image power and present the steady background and tissues with relatively slow variations along the temporal dimension, respectively. Temporal FFT can explore temporal sparsity by gradually eliminating frequency components with negligible magnitude. All other parameters being the same as defined for Eq (2.8).

## 4. Material and methods

The proposed method is tested on two different human abdomen free-breathing Golden-angle-radial DCE-MRI datasets downloaded from "http://cai2r.net/resources/software" [9, 12]. Dataset-1 is a free-breathing human abdomen DCE-MRI dataset acquired from a 3T Siemens Magnetom Prisma scanner. The data acquisition parameters are: 512 readouts, 1144 radial spokes, and 20 receiver coils. Dataset-2 is a free-breathing human abdomen Golden-angle-radial DCE-MRI dataset acquired from 3T Siemens Magnetom Skyra scanner [9, 12]. The data acquisition parameters are: 512 readouts, 1100 radial spokes, and 12 receiver coils. Visual assessment, reconstruction time, convergence error, peak and average values of the dynamic signal curve in the selected region of interest (ROI) of all the frames were used to assess the efficiency, convergence, dynamic performance, and temporal fidelity of the reconstructed DCE-MR images. We calculated Dynamic signal intensity variation in the arterial region for the liver DCE-MRI images to see dynamic contrast variation among different temporal frames. This intensity is used to compare the dynamic contrast performance in all the reconstruction schemes. Reconstruction results obtained from the proposed method are compared with GRASP [1] and XD-GRASP [9] reconstruction. All the experiments were performed using MATLAB (R2021a) on an Intel Core i-7 PC with a 3.2 GHz processor.

In free breathing DCE-MRI, radial k-space data are continuously acquired using the Golden-angle scheme and sorted into time series by grouping an arbitrary number of consecutive spokes into temporal frames. In the proposed method, firstly the acquired golden-angle radial DCE-MRI data are sorted into time frames. 1100 spokes with 512 readouts are acquired at a golden-angle increment, then by selecting 100 spokes in each time frame, generate 11-time frames, i.e., 1100/100 = 11 frames. The Nyquist sampling requirement for this case is 512 * π/2 ≈ 803 projections, corresponding to a simulated acceleration factor (AF) of 8.3, i.e., 803/100 = 8. Similarly, selecting 50 radial spokes in each temporal frame will generate 22 temporal frames (1100/50 = 11) which corresponds to simulated AF of 16 (803/50 = 16).

## 5. Results

This section presents the reconstruction results of the proposed method compared to L+S, GRASP and XD-GRASP reconstructions. Figs 2 and 3 show the reconstruction results of Dataset-1 in venous phase at AF = 8 & 16 respectively. Fig 4 provides comparison of the proposed method, GRASP and XD-GRASP reconstructions in terms of reconstruction time for Dataset-1 for different AFs. The proposed method shows improved reconstruction efficiency

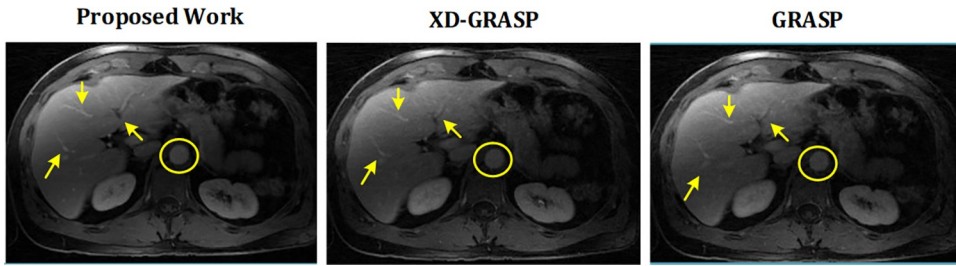

**Fig 2. Reconstruction results of the proposed method compared to GRASP and XD-GRASP reconstruction for human abdomen Dataset-1 at AF = 8 with 11 temporal frames having 96 radial spokes/frame.** The proposed method improved overall image quality, providing better vessel clarity than the GRASP and XD-GRASP reconstructed images shown by arrows.

compared to GRASP and XD-GRASP due to the addition of SC-GROG gridding in L+S reconstruction as shown in Fig 4. XD-GRASP and GRASP has a long reconstruction time due to NUFFT gridding/regridding in NL-CG reconstruction. Fig 4 shows that the proposed method allows approximately 3.5 times improvement in reconstruction time compared to GRASP and 5 times compared to XD-GRASP for Dataset-1 at AF = 8 respectively in our experiments. Signal intensity in the selected region (green circle in Figs 2, 3, 5) is used to compare the dynamic contrast performance in all the reconstruction schemes.

Fig 5 shows the reconstruction results of the proposed method compared to GRASP and XD-GRASP for Dataset-2 at AF = 8 and 16 respectively. Although in the Fig 5, XD-GRASP shows sharper response at some points in the reconstructed images. Anyhow, quantitative experimental results demonstrate that the proposed method consistently outperforms XD-GRASP, providing not only all the necessary information but also a significant improvement in reconstruction time.

Table 4 provides the peak and mean DCE signal intensity values in the three reconstruction schemes for Dataset-1. Temporal TV as sparsity constraint in GRASP and XD-GRASP reduces dynamic contrast resulting in some loss in tissue details. Table 4 shows 8.5% and 6.5% improvement in peak DCE signal intensity value of the proposed method compared to GRASP and XD-GRASP reconstructions respectively for Dataset-1 at AF = 8. Similarly, there is 12.6% and 5.4% improvement in the mean DCE signal intensity value of the proposed method compared to GRASP and XD-GRASP reconstruction respectively for Dataset-1 at AF = 16. This improvement in peak and mean DCE signal intensity is due to temporal FFT as an additional sparsity constraint in the proposed method.

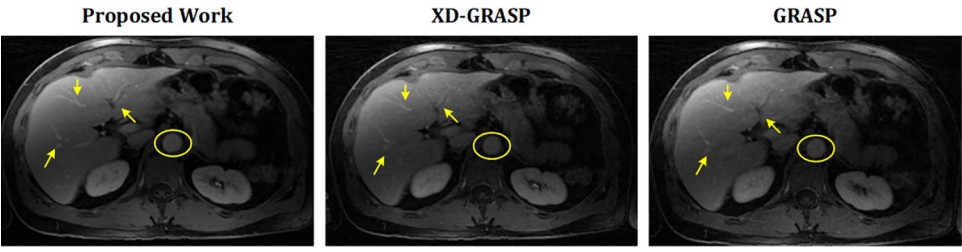

**Fig 3. Reconstruction results of the proposed method compared to XD-GRASP and GRASP reconstructions for human abdomen Dataset-1 at AF = 16 with 22 temporal frames having 50 radial measurements/frame.** MFISTA-VA based optimized *L+S* reconstruction with multiple sparsity constraints (proposed method) provide improved dynamic contrast, and least undersampling artefacts compared to GRASP and XD-GRASP reconstructions.

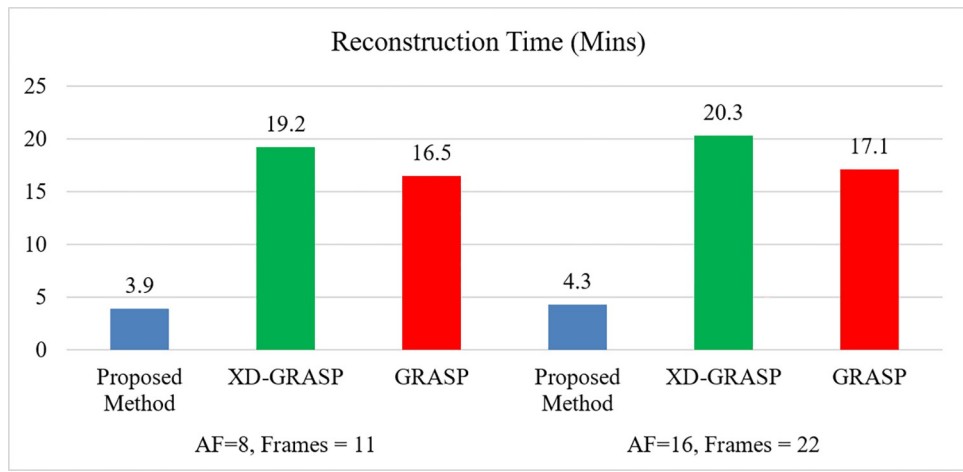

**Fig 4. Comparison of the proposed method, GRASP and XD-GRASP reconstructions in terms of reconstruction time for Dataset-1 for different AFs.** The proposed method shows improved reconstruction efficiency due to incorporation of SC-GROG in L+S reconstruction compared to GRASP and XD-GRASP.

Table 5 provides a comparison of the proposed method, GRASP and XD-GRASP reconstructions in terms of optimization method, gridding techniques, and reconstruction time for Dataset-2 for different AFs. The proposed method provides faster convergence than conventional L+S reconstruction at AF = 8 for Dataset-1 as shown in Fig 6. Fig 7 shows dynamic signal variation among time frames in the selected ROI for different reconstruction schemes at AF = 8 for Dataset-2. The proposed method provides better dynamic contrast than L+S and GRASP reconstructions due to the addition of temporal FFT as an additional sparsity constraint in MFISTA-VA based L+S reconstruction.

## 6. Discussion

This paper proposes a new reconstruction scheme that integrates SC-GROG followed by MFISTA-VA based L+S reconstruction with multiple sparsity constraints for accelerated Golden-angle-radial DCE-MRI with improved temporal resolution and dynamic contrast.

In continuously acquired Golden angle radial MRI data, fully sampled acquisition (with respect to Nyquist criterion), is not performed [1, 2, 6, 9]. Therefore, the evaluation of the proposed method has been performed on the basis of quantitative metrics that do not require a fully sampled ground truth image as a reference. The results of the proposed method have

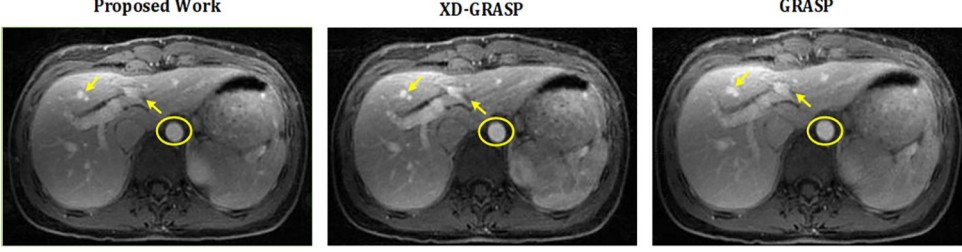

**Fig 5. Reconstruction results of the proposed method compared to GRASP and XD-GRASP reconstructions for human abdomen Dataset-2 at AF = 8 with 11 temporal frames having 96 radial spokes/frame.** The proposed method improved overall image quality, providing better vessel clarity, compared to GRASP and XD-GRASP reconstructed images as shown by arrows.

**Table 4. Dynamic signal variation among different time frames in arterial region in the selected ROI in different reconstruction schemes at AF = 8 and 16 respectively for Dataset-1.** The proposed method provides better dynamic contrast than the other two techniques due to additional sparsity constraint in L+S reconstruction.

| | Peak DCE signal | | Mean DCE signal | |
|---|---|---|---|---|
| AF | 8 | 16 | 8 | 16 |
| GRASP | 0.5078 | 0.4848 | 0.3527 | 0.2740 |
| XD-GRASP | 0.5155 | 0.4970 | 0.3774 | 0.2988 |
| Proposed Method | 0.5510 | 0.5324 | 0.3978 | 0.3140 |

been compared with the results of two conventional methods i.e. GRASP and XD-GRASP. Among these two methods, GRASP does not employ binning for reconstructing the dynamic contrast-enhanced data. So the comparison of the proposed method with GRASP corresponds to a comparison with non-binned data. We also compared our method with conventional L+S for comparison of reconstruction time and convergence analysis.

Experimental results in Figs 2, 3 and 5 show quality improvement in the MR images obtained from the proposed method at different AFs compared to the GRASP and XD-GRASP reconstructions. The results show that the left, middle, and right hepatic veins, stomach, and pancreas are a bit blurred in the GRASP and XD-GRASP reconstructions, while they have better resolution in the images obtained from the proposed method.

Conventional L+S uses ISTA to obtain an optimal solution. In this work, we used MFISTA-VA to solve the L+S optimization problem which improves the convergence of Conventional L+S reconstruction. MFISTA-VA uses variable step size during iterative CS reconstruction which increases convergence of MFISTA-VA compared to FISTA which uses constant step size. Proposed method provides faster convergence over conventional L+S reconstruction as shown in Fig 6. The results show that the proposed method provides up to 5× times and 3× time improvement in the reconstruction time compared to GRASP and XD-GRASP reconstructions for Dataset-2 at AF = 8 respectively.

GRASP and XD-GRASP use NUFFT gridding [36] to convert non-Cartesian data to a Cartesian grid before NL-CG reconstruction. NL-CG reconstruction is computationally expensive as it requires repeated NUFFT gridding/degridding to maintain data consistency during iterative CS reconstruction. SC-GROG is computationally efficient compared to NUFFT gridding due to following factors, (i) it does not require some extra computations, e.g., density compensation function (DCF), convolution kernel and windows sizes and (ii) it does not require gridding and de-gridding (to maintain data consistency) during iterative CS reconstruction. Proposed method uses SC-GROG to convert golden-angle radial data to Cartesian grid before

**Table 5. Comparison of the proposed method, standard *L+S*, GRASP and XD-GRASP reconstructions in terms of optimization method, gridding technique, and reconstruction time for Dataset-2 for different AFs.** The proposed method shows improved reconstruction efficiency compared to *L+S*, GRASP and XD-GRASP reconstruction due to the addition of SC-GROG gridding and MFISTA-VA based optimization in standard *L+S* reconstruction.

| Reconstruction Method | AF | Optimization Method | Gridding Method | Number of Frames | Reconstruction Time (Mins) |
|---|---|---|---|---|---|
| Proposed Method | 8 | MFISTA-VA based L+S Reconstruction | GROG | 11 | 4.5 |
| | 16 | | | 22 | 5.1 |
| Standard *L+S* | 8 | ISTA based L+S Reconstruction | NUFFT | 11 | 18.5 |
| | 16 | | | 22 | 19.3 |
| XD-GRASP | 8 | NL-CG | NUFFT | 11 | 21.3 |
| | 16 | | | 22 | 22.7 |
| GRASP | 8 | NL-CG | NUFFT | 11 | 15.5 |
| | 16 | | | 22 | 16.3 |

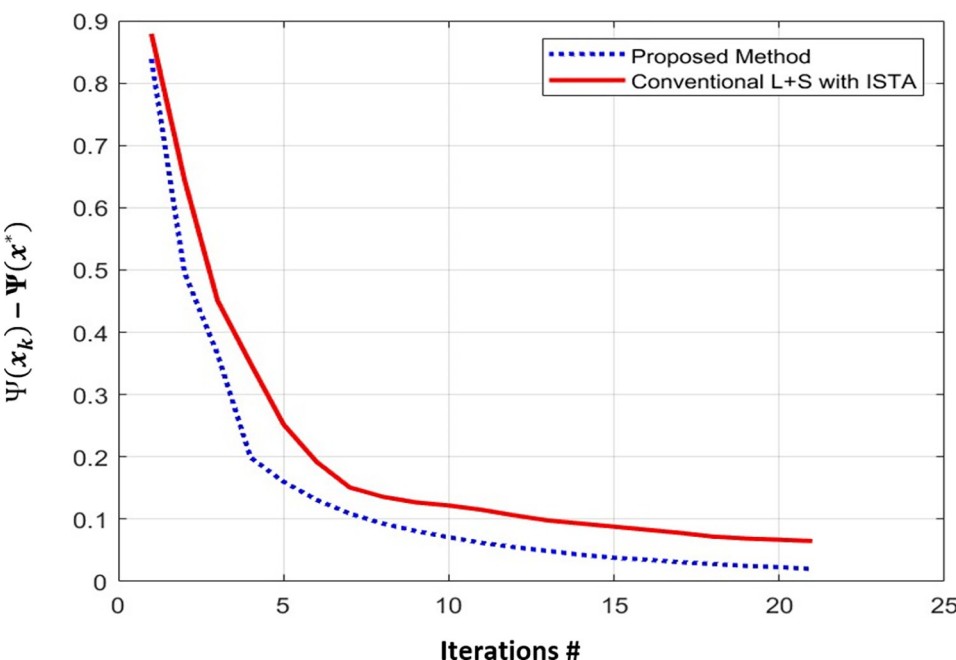

**Fig 6. Curves showing the cost function $\Psi(x_k)-\Psi(x^*)$ over # of iterations.** The proposed method provides faster convergence than conventional L+S reconstruction at AF = 8 for Dataset-1.

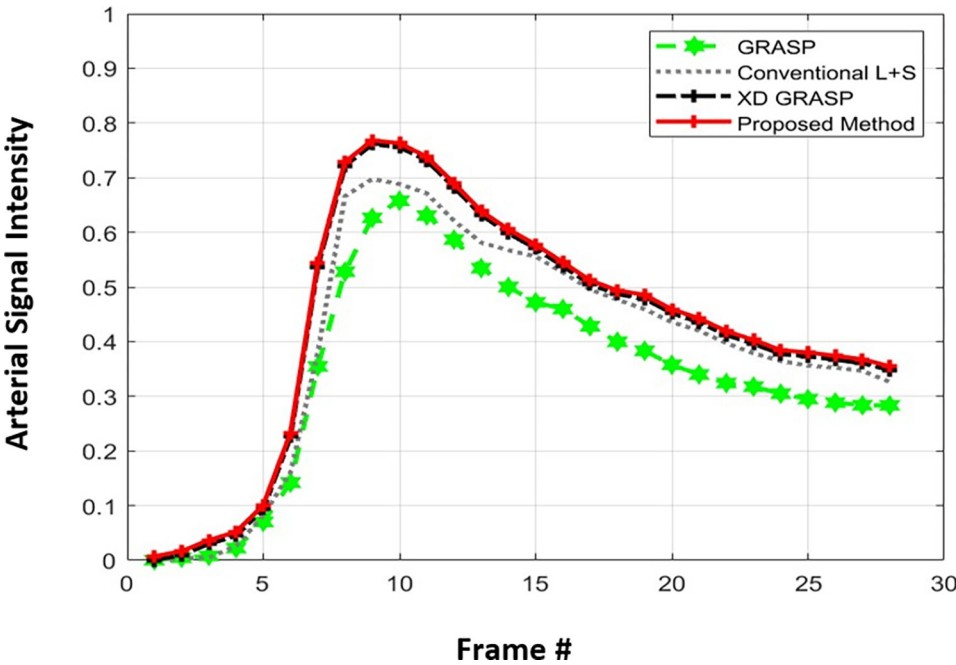

**Fig 7. Dynamic signal variation among time frames in the selected ROI for different reconstruction schemes at AF = 8 for Dataset-2.** The proposed method provides better dynamic contrast than GRASP, L+S and XD-GRASP reconstructions due to the addition of temporal FFT as an additional sparsity constraint in MFISTA-VA based L+S reconstruction.

MFISTA-VA based L+S reconstruction. Incorporation of SC-GROG gridding in the proposed method increases reconstruction efficiency as shown in Fig 4 and Table 5.

## 7. Conclusion

This paper presents a novel reconstruction scheme for free-breathing Golden-angle-radial DCE-MRI with improved dynamic contrast better time efficiency and image quality by integrating SC-GROG gridding followed by MFISTA-VA based optimized *L+S* reconstruction with multiple sparsity constraint. By integrating SC-GROG gridding, the reconstruction process is made more time-efficient. Unlike NUFFT gridding, SC-GROG gridding eliminates the need for time-consuming gridding and regridding steps during iterative reconstruction. The use of multiple sparsity constraints in the reconstruction process enhances dynamic contrast, enabling the capture of finer tissue details and reducing artifacts. The proposed method also incorporates MFISTA-VA, an optimized algorithm that leverages variable acceleration. This results in a decrease in the reconstruction time by accelerating the convergence of the reconstruction process compared to traditional methods like FISTA and ISTA. Overall, the proposed method provides a significant improvement in time efficiency, dynamic contrast, and image quality when compared to conventional GRASP and XD-GRASP techniques. In the future work, parallel computing using GPU (graphics processing unit) may be used to further improve the computational speed of SC-GROG gridding.

## Author Contributions

**Conceptualization:** Faisal Najeeb, Kashif Amjad, Irfan Ullah, Hammad Omer.

**Data curation:** Faisal Najeeb.

**Formal analysis:** Faisal Najeeb, Irfan Ullah.

**Investigation:** Faisal Najeeb, Hammad Omer.

**Methodology:** Faisal Najeeb.

**Resources:** Faisal Najeeb, Irfan Ullah.

**Software:** Faisal Najeeb.

**Supervision:** Kashif Amjad, Hammad Omer.

**Validation:** Faisal Najeeb.

**Visualization:** Faisal Najeeb, Irfan Ullah.

**Writing – original draft:** Faisal Najeeb.

**Writing – review & editing:** Kashif Amjad, Hammad Omer.

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
