## [Decision Letter · Decision Letter 0]

2 Jul 2024

PONE-D-24-06314SC-GROG followed by L+S reconstruction with multiple sparsity constraints for accelerated Golden-angle-radial DCE-MRIPLOS ONE

Dear Dr. Amjad,

Thank you for submitting your manuscript to PLOS ONE. After careful consideration, we feel that it has merit but does not fully meet PLOS ONE’s publication criteria as it currently stands. Therefore, we invite you to submit a revised version of the manuscript that addresses the points raised during the review process. The manuscript was reviewed by two expert reviewers. Both found the work interesting and of value, but the presentation needs to be improved. Including improving the language and style (language editing is this recommended).. 

We look forward to receiving your revised manuscript.

Kind regards,

Peter Lundberg

Academic Editor

PLOS ONE

Reviewers' comments:

Reviewer's Responses to Questions

**Comments to the Author**

1. Is the manuscript technically sound, and do the data support the conclusions?

Reviewer #1: Yes

Reviewer #2: Yes

2. Has the statistical analysis been performed appropriately and rigorously? 

Reviewer #1: Yes

Reviewer #2: N/A

3. Have the authors made all data underlying the findings in their manuscript fully available?

Reviewer #1: Yes

Reviewer #2: Yes

4. Is the manuscript presented in an intelligible fashion and written in standard English?

Reviewer #1: Yes

Reviewer #2: Yes

5. Review Comments to the Author

Reviewer #1: This work aims to improve the dynamic contrast and reduce the reconstruction

time of CS-based image reconstruction problems in DCE-MRI.

The authors compare their proposed method with two other.

Especially, I like Figure 1 and Table 5! They are educational!

Below I mix comments regarding grammar with more important comments.

Abstract, row 5 from bottom

and (iii) reconstruction time.

=>

, (iii) reconstruction time, and (iv) image examples.

1. Introduction, row 11

In GRASP, data was

=>

In GRASP, data are

1. Introduction, row 12

contrast-enhanced images were reconstructed

=>

contrast-enhanced images are reconstructed

1. Introduction, row 26

Although GRASP based

=>

Although GRASP-based

1. Introduction, row 27

performance, however

=>

performance, however,

1. Introduction, row 29

NUFFT ridding/degrading

=>

NUFFT gridding/degrading

1. Introduction, row 12 from the bottom

A point is missing.

1. Introduction, row 8 from the bottom

L+S

=>

L+S reconstruction

1. Introduction, row 5 from the bottom

...work also uses SC-GROG ...

You have already mentioned that!

2.1 ITERATIVE SOFT THRESHOLDING ALGORITHMS, row 1

defined over complex

=>

defined over the complex

2.1 ITERATIVE SOFT THRESHOLDING ALGORITHMS, row 4

Here h(x) represents sparsity

=>

Here h(x) represents the sparsity

2.1 ITERATIVE SOFT THRESHOLDING ALGORITHMS, row 5

Please explain y!

2.1 ITERATIVE SOFT THRESHOLDING ALGORITHMS, row 5-6

Lipschitz constant, || || shows

=>

the Lipschitz constant, || || denotes

2.1 ITERATIVE SOFT THRESHOLDING ALGORITHMS, row 8-9

problem to reconstruct MR images , given the partially acquired k-space data d,

=>

problems, reconstruction of an MR image x,

2.1 ITERATIVE SOFT THRESHOLDING ALGORITHMS, row 11

In equation (2.2)

=>

In equation (2.2),

2.1 ITERATIVE SOFT THRESHOLDING ALGORITHMS, row 11

vector

=>

the vector

2.1 ITERATIVE SOFT THRESHOLDING ALGORITHMS, row 14

sparsity

=>

the sparsity

2.1 ITERATIVE SOFT THRESHOLDING ALGORITHMS, row 19

method Proximal-gradient

=>

methods, a Proximal-gradient

2.1 ITERATIVE SOFT THRESHOLDING ALGORITHMS, row 25

... were proposed almost a decade ...

You repeat yourself!

2.1 ITERATIVE SOFT THRESHOLDING ALGORITHMS, row 17 from the bottom

major condition

=>

then a major condition

2.1 ITERATIVE SOFT THRESHOLDING ALGORITHMS, row 14 from the bottom

here x* represents minimizer of the Psi, x_k denotes k_th iteration in M-FISTA with constant parameter W in

=>

Here x* represents the minimizer of Psi, x_k denotes the k_th iteration in M-FISTA with the constant parameter W as

2.1 ITERATIVE SOFT THRESHOLDING ALGORITHMS, row 13 from the bottom

satisfy convergence

=>

satisfy the convergence

2.2 MR RECONSTRUCTION USING ROBUST MATRIX DECOMPOSITION, row 2

under sampled

=>

undersampled

2.2 MR RECONSTRUCTION USING ROBUST MATRIX DECOMPOSITION, row 3

shows

=>

shows a

2.2 MR RECONSTRUCTION USING ROBUST MATRIX DECOMPOSITION, row 9

less

=>

small

2.2 MR RECONSTRUCTION USING ROBUST MATRIX DECOMPOSITION, row 1 under TABLE 1

multi-coil under sampled dynamic MRI data, L+S decomposition

=>

multi-coil undersampled dynamic MRI data, the L+S decomposition

2.2 MR RECONSTRUCTION USING ROBUST MATRIX DECOMPOSITION, row 7 under TABLE 1

regularization parameters for L

=>

regularization parameters for the L

2.2 MR RECONSTRUCTION USING ROBUST MATRIX DECOMPOSITION, row 10 under TABLE 1

ISTA/shrinkage operator

=>

The ISTA/shrinkage operator

2.2 MR RECONSTRUCTION USING ROBUST MATRIX DECOMPOSITION, row 24 under TABLE 1

residual

=>

the residual

2.2 MR RECONSTRUCTION USING ROBUST MATRIX DECOMPOSITION, row 26 under TABLE 1

proximal

=>

the proximal

2.2 MR RECONSTRUCTION USING ROBUST MATRIX DECOMPOSITION, row 29 under TABLE 1

sum of nuclear

=>

the sum of the nuclear

3. PROPOSED METHOD, row 4

Flow-graph

=>

A flowchart

3. PROPOSED METHOD, row 5

image domain

=>

the image domain

Equation (3.1)

G.

=>

G

1 row below Equation (3.1)

G shows GROG

=>

G is the GROG

2 rows below Equation (3.2)

shows sparsity

=>

is the sparsity

3 rows below Equation (3.2)

You write equation (2.7) but I think that you mean equation (2.8).

4. MATERIALS AND METHODS, row 3

DCE-MRI data acquired from

=>

DCE-MRI dataset acquired from a

4. MATERIALS AND METHODS, row 5

data

=>

dataset

4. MATERIALS AND METHODS, row 8

dynamic

=>

the dynamic

5. RESULTS, row 2

You write about figures 2 and 3. Describe also what Figure 5 shows!

Figure 2, 3, 5. IMPORTANT!

You need to help the reader!

The arrows should be numbered and the respective artifact at each arrow

should be explained in the text.

FIGURE 2, figure text row 3

vessel clarity, and

=>

vessel clarity than the

FIGURE 6, figure text row 2

fast convergence over conventional

=>

faster convergence than conventional

6. DISCUSSION, row 9

XD-GRASP

=>

GRASP and XD-GRASP

6. DISCUSSION, row 13-15

allows up to 5x times and 3x time improvement in the reconstruction time compared to XD-GRASP and GRASP reconstructions respectively for Dataset-2 at AF=8.

=>

provides up to 5 and 3 times faster reconstruction time compared to XD-GRASP and GRASP reconstructions

for dataset 2 at AF=8, respectively.

6. DISCUSSION

Are there any limitations with the new suggested method?

Figure 7

========

This figure must also be referenced in the text!

Reviewer #2: The article discusses advancements in dynamic contrast-enhanced magnetic resonance imaging (DCE-MRI), focusing on improving the GRASP (Golden-angle-radial Sparse Parallel MRI) reconstruction method. While GRASP leverages advanced sampling patterns, parallel MRI, and compressed sensing for accelerated, free-breathing DCE-MRI reconstruction, it faces limitations such as temporal averaging effects and high computational demands. To address these issues, the paper proposes a novel approach that integrates SC-GROG gridding with low-rank plus sparse (L+S) reconstruction and multiple sparsity constraints, using Monotone FISTA with variable acceleration (MFISTA-VA) for optimization. Tested on two 3T free-breathing in-vivo DCE-MRI datasets, the proposed method significantly improves reconstruction time and dynamic contrast, outperforming conventional methods like GRASP and XD-GRASP, and achieves faster convergence with MFISTA-VA in L+S reconstruction.

After the review I have following comments to make this submission more suitable for the journal

Major Comment

1. How MFISTA-VA is computationally better than the soft thresholding used in 1, 2, 6 and 9?

2. Can you explain the difference between this article? Shahzadi, Iram, et al. "Golden-angle radial sparse parallel MR image reconstruction using SC-GROG followed by iterative soft thresholding." Applied Magnetic Resonance 50 (2019): 977-988.

3. It seems your group has done quite a lot of work on GROG, GRASP and Compresses Sensing based reconstructions, it will be better if you can provide a comparison of results with this article as well.

4. Section 2.2 contains a lot of information from the original paper by Ricardo Otazo about L+S Decomposition Model. It is better to reduce this section to show how MFISTA-VA is applied for L+S Model.

5. Which algorithm was used to estimate the sensitivity maps?

6. Expression 3.2 there is no explanation how TemporalTV and and Temporal FFT is applied on sparse data separately.

7. “Table 4 shows that the peak DCE signal was degraded by 8.5% and 6.5% ….. ” Degraded in comparison of what ? Did you use any reference image? How about adding a non-binned image for comparison ?

8. In discussion section there is no mention of the reason why this SC GROG based method performs better than conventional methods. Which is very important for this section.

Minor Comments

1. Introduction: Paragraph 1: “physiological movement (e.g., respiratory, and cardiac motion) introduces blurring artefacts”, these movements don’t only add up to the blurring artifacts, there are more artifacts linked with these movements.

2. Introduction: Paragraph 3: “Temporal TV can suppress all the temporal variations”, how the spatial resolution is tackled?

3. Introduction: Paragraph 5: Repetitive sentence. “To generate good quality DCE MR images, intensity variation due to contrast agent and the respiratory motion must be separated from each other before image reconstruction”.

4. Theory Section: 2.1 Iterative Soft Thresholding Methods. It is good to talk about the thresholding methods, while on the other hand it is better to keep it precise to make it more readable. It is better to focus only on MFISTA-VA which is used in this paper.

5. “Tables 4 provides the peak and mean DCE signal intensity values in the three reconstruction schemes for Dataset-1 and Dataset-2 respectively”, are you comparing the complete image or just the sparse images ? How it behaves if you just compare sparse images?

6. Figure 4 is a bit confusing. It is not clear if you are talking about a 11 frames reconstruction, or 22 frames reconstruction in total or you are just showing results for these two particular frames.

7. There is not explanation of Figure 6 and Figure 7 in Results Section. And the plots are very blurry, and not suitable to be included in the paper in current form.

8. Figure 6 and 7, needs improvement, as they both need all the methods mentioned for comparison in the results section.

Formatting: There are quite a few grammatical errors, like

“NL-CG reconstruction which requires repeated NUFFT ridding/degrading to maintain”

“Where represents the under-sampled GROG-gridded Cartesian.”, complete this sentence.

The placement of figures and tables needs to be corrected. Like Figure 4, comes before table 4, while you talk about it later in the text. Same for the Tables with pseudo codes for algorithms.

6. PLOS authors have the option to publish the peer review history of their article (what does this mean?). If published, this will include your full peer review and any attached files.

Reviewer #1: **Yes: **Maria Magnusson

Reviewer #2: No

---

## [Author Response · Author response to Decision Letter 0]

30 Aug 2024

We are thankful to the reviewers for their deep and thorough review of our paper “SC-GROG 

followed by L+S reconstruction with multiple sparsity constraints for accelerated Golden angle-radial DCE-MRI”. We have revised the manuscript in the light of their useful suggestions 

and comments. We hope that this revision has improved the paper to a level of their satisfaction. 

Number-wise answers to the specific comments/suggestions/queries are attached in the response to reviewers file in the submitted revised version.

---

## [Decision Letter · Decision Letter 1]

5 Nov 2024

PONE-D-24-06314R1SC-GROG followed by L+S reconstruction with multiple sparsity constraints for accelerated Golden-angle-radial DCE-MRIPLOS ONE

Dear Dr. Amjad,

Thank you for submitting your manuscript to PLOS ONE. After careful consideration, we feel that it has merit but does not fully meet PLOS ONE’s publication criteria as it currently stands. Therefore, we invite you to submit a revised version of the manuscript that addresses the points raised during the review process. Please submit your revised manuscript by Dec 20 2024 11:59PM. If you will need more time than this to complete your revisions, please reply to this message or contact the journal office at plosone@plos.org. Please include the following items when submitting your revised manuscript:A rebuttal letter that responds to each point raised by the academic editor and reviewer(s). You should upload this letter as a separate file labeled 'Response to Reviewers'.A marked-up copy of your manuscript that highlights changes made to the original version. You should upload this as a separate file labeled 'Revised Manuscript with Track Changes'.An unmarked version of your revised paper without tracked changes. You should upload this as a separate file labeled 'Manuscript'.If applicable, we recommend that you deposit your laboratory protocols in protocols.io to enhance the reproducibility of your results. Protocols.io assigns your protocol its own identifier (DOI) so that it can be cited independently in the future. For instructions see: https://journals.plos.org/plosone/s/submission-guidelines#loc-laboratory-protocols. Additionally, PLOS ONE offers an option for publishing peer-reviewed Lab Protocol articles, which describe protocols hosted on protocols.io. Read more information on sharing protocols at https://plos.org/protocols?utm_medium=editorial-email&utm_source=authorletters&utm_campaign=protocols.

We look forward to receiving your revised manuscript.

Kind regards,

Peter Lundberg

Academic Editor

PLOS ONE

Journal Requirements:

**Additional Editor Comments:**

The manuscript is in a much better shape, which is appreciated, and it is almost ready for publication. However, the reviewers have some additional comments that needs to be considered.

Reviewers' comments:

Reviewer's Responses to Questions

**Comments to the Author**

1. If the authors have adequately addressed your comments raised in a previous round of review and you feel that this manuscript is now acceptable for publication, you may indicate that here to bypass the “Comments to the Author” section, enter your conflict of interest statement in the “Confidential to Editor” section, and submit your "Accept" recommendation.

Reviewer #1: All comments have been addressed

Reviewer #2: (No Response)

2. Is the manuscript technically sound, and do the data support the conclusions?

Reviewer #1: Yes

Reviewer #2: Yes

3. Has the statistical analysis been performed appropriately and rigorously? 

Reviewer #1: Yes

Reviewer #2: Yes

4. Have the authors made all data underlying the findings in their manuscript fully available?

Reviewer #1: Yes

Reviewer #2: Yes

5. Is the manuscript presented in an intelligible fashion and written in standard English?

Reviewer #1: (No Response)

Reviewer #2: Yes

6. Review Comments to the Author

Reviewer #1: The following text was below eq. (2.7) in the previous submission. Now it is missing.

"Here, the sparse matrix S models the grossly corrupted noise/outlier information

which is assumed to be a very small fraction of the acquired MRI data. Lambda_T

represents regularization parameters for S component."

IMPORTANT:

In Figure 5, I think the XD-GRASP image looks sharper and more detailed than

the proposed work. Can you elaborate on that?

Reviewer #2: For My previous Major Comment "Table 4 shows that the peak DCE signal was degraded by 8.5% and 6.5% ….. ” Degraded in comparison of what? Did you use any reference image? How about adding a non-binned image for comparison?"

If there is no reference data, the claim of improvement becomes less compelling. I strongly recommend comparing the results with non-binned data, as it would provide a clearer benchmark for evaluating dynamic contrast performance. Including such a comparison would significantly strengthen your conclusions, or alternatively, please provide a clear rationale for why non-binned data cannot be used.

7. PLOS authors have the option to publish the peer review history of their article (what does this mean?). If published, this will include your full peer review and any attached files.

Reviewer #1: **Yes: **Maria Magnusson

Reviewer #2: No

---

## [Author Response · Author response to Decision Letter 1]

20 Dec 2024

Dear Editor , we uploaded a seperate response to reviewers file.

---

## [Editor Report · Decision Letter 2]

10 Jan 2025

SC-GROG followed by L+S reconstruction with multiple sparsity constraints for accelerated Golden-angle-radial DCE-MRI

PONE-D-24-06314R2

Dear Dr. Amjad,

We’re pleased to inform you that your manuscript has been judged scientifically suitable for publication and will be formally accepted for publication once it meets all outstanding technical requirements.

Kind regards,

Peter Lundberg

Academic Editor

PLOS ONE
---

## [Editor Report · Acceptance letter]

24 Jan 2025

PONE-D-24-06314R2 

PLOS ONE

Dear Dr. Amjad, 

I'm pleased to inform you that your manuscript has been deemed suitable for publication in PLOS ONE. Congratulations! Your manuscript is now being handed over to our production team.

Kind regards, 

on behalf of

Professor Peter Lundberg 

Academic Editor

PLOS ONE